# SARS-CoV-2 evolution balances conflicting roles of N protein phosphorylation

**Abdullah M. Syed**[1][ꝏ], **Alison Ciling**[1][ꝏ], **Irene P. Chen**[1,2], **Christopher R. Carlson**[3], **Armin N. Adly**[3], **Hannah S. Martin**[4], **Taha Y. Taha**[1], **Mir M. Khalid**[1], **Nathan Price**[1], **Mehdi Bouhaddou**[1,5,6,7,8,9], **Manisha R. Ummadi**[1,7,8,9], **Jack M. Moen**[1,7,8,9], **Nevan J. Krogan**[1,7,8,9,10], **David O. Morgan**[3], **Melanie Ott**[1,2], **Jennifer A. Doudna**[1,11,12,13,14,15]*

1 Gladstone Institutes, San Francisco, California, United States of America, 2 Department of Medicine, University of California, San Francisco, California, United States of America, 3 Department of Physiology, University of California, San Francisco, California, United States of America, 4 Department of Chemistry, University of California, Berkeley, California, United States of America, 5 Department of Microbiology, Immunology, and Molecular Genetics (MIMG), UCLA, Los Angeles, California, United States of America, 6 Institute for Quantitative and Computational Biosciences (QCBio), UCLA, Los Angeles, California, United States of America, 7 Quantitative Biosciences Institute (QBI), University of California, San Francisco, California, United States of America, 8 Quantitative Biosciences Institute (QBI) COVID-19 Research Group (QCRG), San Francisco, California, United States of America, 9 Department of Cellular and Molecular Pharmacology, University of California, San Francisco, California, United States of America, 10 Department of Bioengineering and Therapeutic Sciences, University of California, San Francisco, California, United States of America, 11 Department of Molecular and Cell Biology, University of California, Berkeley, California, United States of America, 12 Howard Hughes Medical Institute, University of California, Berkeley, California, United States of America, 13 Innovative Genomics Institute, University of California, Berkeley, California, United States of America, 14 California Institute for Quantitative Biosciences (QB3), University of California, Berkeley, California, United States of America, 15 MBIB Division, Lawrence Berkeley National Laboratory, Berkeley, California, United States of America

ꝏ These authors contributed equally to this work.

* doudna@berkeley.edu

**Data Availability Statement:** All relevant data are within the manuscript and its Supporting Information files.

## Abstract

All lineages of SARS-CoV-2, the coronavirus responsible for the COVID-19 pandemic, contain mutations between amino acids 199 and 205 in the nucleocapsid (N) protein that are associated with increased infectivity. The effects of these mutations have been difficult to determine because N protein contributes to both viral replication and viral particle assembly during infection. Here, we used single-cycle infection and virus-like particle assays to show that N protein phosphorylation has opposing effects on viral assembly and genome replication. Ancestral SARS-CoV-2 N protein is densely phosphorylated, leading to higher levels of genome replication but 10-fold lower particle assembly compared to evolved variants with low N protein phosphorylation, such as Delta (N:R203M), Iota (N:S202R), and B.1.2 (N:P199L). A new open reading frame encoding a truncated N protein called N*, which occurs in the B.1.1 lineage and subsequent lineages of the Alpha, Gamma, and Omicron variants, supports high levels of both assembly and replication. Our findings help explain the enhanced fitness of viral variants of concern and a potential avenue for continued viral selection.

**Funding:** This project was funded by grants from the National Institutes of Health (R21-AI59666 to JAD; R35-GM118053 to DOM; U19AI135990 and U19AI135972 to NJK) and by support from the Howard Hughes Medical Institute and the Gladstone Institutes to JAD. The funders had no role in study design, data collection and analysis, decision to publish, or preparation of the manuscript.

**Competing interests:** AMS and JAD are co-inventors on a patent application filed by Gladstone Institutes and University of California on the generation of SARS-CoV-2 VLPs. TYT and MO are inventors on a patent application filed by the Gladstone Institutes that covers the use of pGLUE to generate SARS-CoV-2 infectious clones and replicons. The NJK Laboratory has received research support from Vir Biotechnology, F. Hoffmann-La Roche, and Rezo Therapeutics. NJK has a financially compensated consulting agreement with Maze Therapeutics. NJK is the President and is on the Board of Directors of Rezo Therapeutics, and he is a shareholder in Tenaya Therapeutics, Maze Therapeutics, Rezo Therapeutics, GEn1E Lifesciences, and Interline Therapeutics. All other authors declare no competing interests.

## Author summary

The COVID-19 pandemic was caused by SARS-CoV-2 and is characterized by waves of evolved viral variants with mutations across the viral genome. Nucleocapsid (N) is a viral protein that is critical for viral RNA packaging into particles and for viral replication in cells. Over the pandemic, N has acquired several mutations between amino acids 199 and 205 that are associated with increased infectivity. The mechanism underlying these mutations has been difficult to elucidate given the multiple roles of N protein in the viral life-cycle. Here, we developed methods to separately determine the effect of mutations on viral particle assembly vs viral RNA replication. We found that the mutations between amino acids 199 and 205 affect N protein phosphorylation and have opposite effects on viral assembly and genome replication. Ancestral SARS-CoV-2 N protein is densely phosphorylated, leading to higher levels of genome replication but 10-fold lower particle assembly compared to evolved variants with low N protein phosphorylation. One of the mutations found in the Alpha, Gamma, and Omicron variants of concern results in expression of a new, truncated N protein called N* that supports high levels of both assembly and replication. Our findings help explain the enhanced fitness of viral variants of concern and a potential avenue for continued viral selection.

## Introduction

Understanding the impact of mutations in SARS-CoV-2, the virus responsible for the COVID-19 pandemic, provides insights into its transmission, pathogenesis and evolution. SARS-CoV-2 encodes 16 non-structural proteins (nsp1-16), four structural proteins (S, N, M and E) and up to 11 accessory proteins. The Spike (S) protein mediates viral entry and has been a primary research focus due to the role of its mutations in evading host immune responses and since it can be studied easily using pseudovirus assays. However, a high density of mutations in circulating variants also occur in the Nucleocapsid (N) protein, suggesting that it plays a significant role in viral fitness and evolution. Indeed, previous studies using SARS-CoV-2 virus-like particles and infectious clone assays revealed a central role for N protein mutations in enhanced viral particle assembly and fitness [1–3].

The effects of mutations in the N protein are hard to dissect due to its pivotal role in both viral genome replication and particle assembly. N protein forms a ribonucleoprotein complex with the viral RNA genome, facilitating replication, transcription, and packaging of the viral genome into nascent virions. The N protein contains an N-terminal domain (NTD), a C-terminal domain (CTD) and three intrinsically disordered regions (N-arm, linker, C-tail). A conserved central serine/arginine-rich (SR) motif within the central linker region is subject to phosphorylation, predominantly by the host kinase glycogen synthase kinase 3 (GSK-3) [3–5]. This motif contains more than 10 phosphorylation sites located in a dense cluster and is known to regulate multiple functions of N protein including phase separation [6–9], RNA packaging [10,11], subgenomic RNA synthesis [12], regulation of interferon signaling [13–15], and nuclear/cytosolic localization [16]. However, the effects of N protein phosphorylation on the different stages of the SARS-CoV-2 viral life cycle remain unclear.

Here, we show that SARS-CoV-2 N protein phosphorylation creates an inherent conflict between viral genome replication and assembly. We dissected the effects of the SR motif's phosphorylation on critical steps of the viral life cycle by developing and utilizing assays that isolate viral assembly and genome replication using virus-like particles (VLPs) and single cycle infectious particles (SCIPs), respectively. We find that phosphorylation of the SR motif inhibits

assembly but improves genome replication, suggesting an evolutionary trade-off between these functions. Characterization of a truncated version of N, termed N*, found in newer SARS-CoV-2 variants indicates that these functions could in principle be carried out by two separate proteins. Nonetheless, it is not clear that currently circulating variants exploit this protein for this purpose. Our findings show how viral evolution could resolve the conflict between these two opposing functions of N protein and provide a window into its potential future evolutionary trajectory.

## Results

### Mutations in N protein inhibit its sequential phosphorylation

The phosphorylation of the SR motif in SARS-CoV-2 N protein is hypothesized to commence with a Serine/arginine-Rich Splicing Factor protein kinase (SRPK) or another kinase phosphorylating at positions 188 and 206 (Fig 1A) [4]. Following these priming events, GSK-3 phosphorylates sequentially at every fourth residue upstream of the priming sites [4,17,18]. Casein kinase 1 (CK1) is then thought to modify additional sites downstream of the GSK-3 sites [4]. Within circulating variants, position 203 and neighboring residues are prone to mutations, with independently arising amino acid changes at these locations observed in all variants of concern (Fig 1B). In our previous study, we determined that mutations P199L, S202R, R203M and R203K significantly enhanced viral assembly and fitness over 10-fold [1]. Taking into account the proximity of these mutations to key GSK-3 phosphorylation sites, we explored whether these mutations would alter the extent of N protein phosphorylation.

To assess phosphorylation, we used two complementary approaches. First, we expressed and purified four SARS-CoV-2 N protein variants previously shown to enhance viral assembly, each containing a single amino acid change relative to ancestral protein: P199L, S202R, R203M, and R203K [1]. We also purified the ancestral protein, two single-mutation N proteins as negative controls (G204R and M234I), and two single mutation N proteins targeting the GSK-3 priming sites as positive controls (S188A and S206A). We analyzed the phosphorylation of these mutant proteins biochemically with purified kinase enzymes SRPK, GSK3, and CK1, plus $^{32}$P-labeled ATP (Fig 1C and 1D). We found that R203M, present in the SARS-CoV-2 Delta variant, and R203K and R203K/G204R, found in all lineage B.1.1 variants, reduced phosphorylation in this assay to a level equivalent to S206A, suggesting a substantial decrease in N protein phosphorylation in these variants (Figs 1C, 1D, and S1A).

To assess phosphorylation of N protein in human cells, we exploited the difference in migration of phosphorylated and unmodified N proteins on a 7.5% SDS-PAGE gel based on previous reports [5,8]. Treatment with CHIR98014 (a potent and specific GSK-3 inhibitor) as well as mutations in both priming sites (S188A/S206A), resulted in a faster migrating species consistent with an unphosphorylated N protein (Figs 1E and S1B). We also used a phosphate affinity gel (Phos-Tag[19]) to verify the phosphorylation state of N protein but observed a smear instead of distinct bands for most N protein variants likely due to the high number of phosphorylation sites present in the SR region (S1C Fig). Using a standard SDS-PAGE gel, we observed reduced N protein phosphorylation for P199L, S202R, R203M and R203K single point mutations, but not G204R or M234I (Figs 1E and S1D). These observations indicate that SARS-CoV-2 variants have independently evolved multiple N protein mutations that reduce phosphorylation compared to the ancestral variant.

### N protein phosphorylation inhibits viral particle assembly

Next, we tested whether decreased phosphorylation of N protein corresponds to improved viral assembly. We generated virus-like particles (VLPs) by co-transfecting SARS-CoV-2

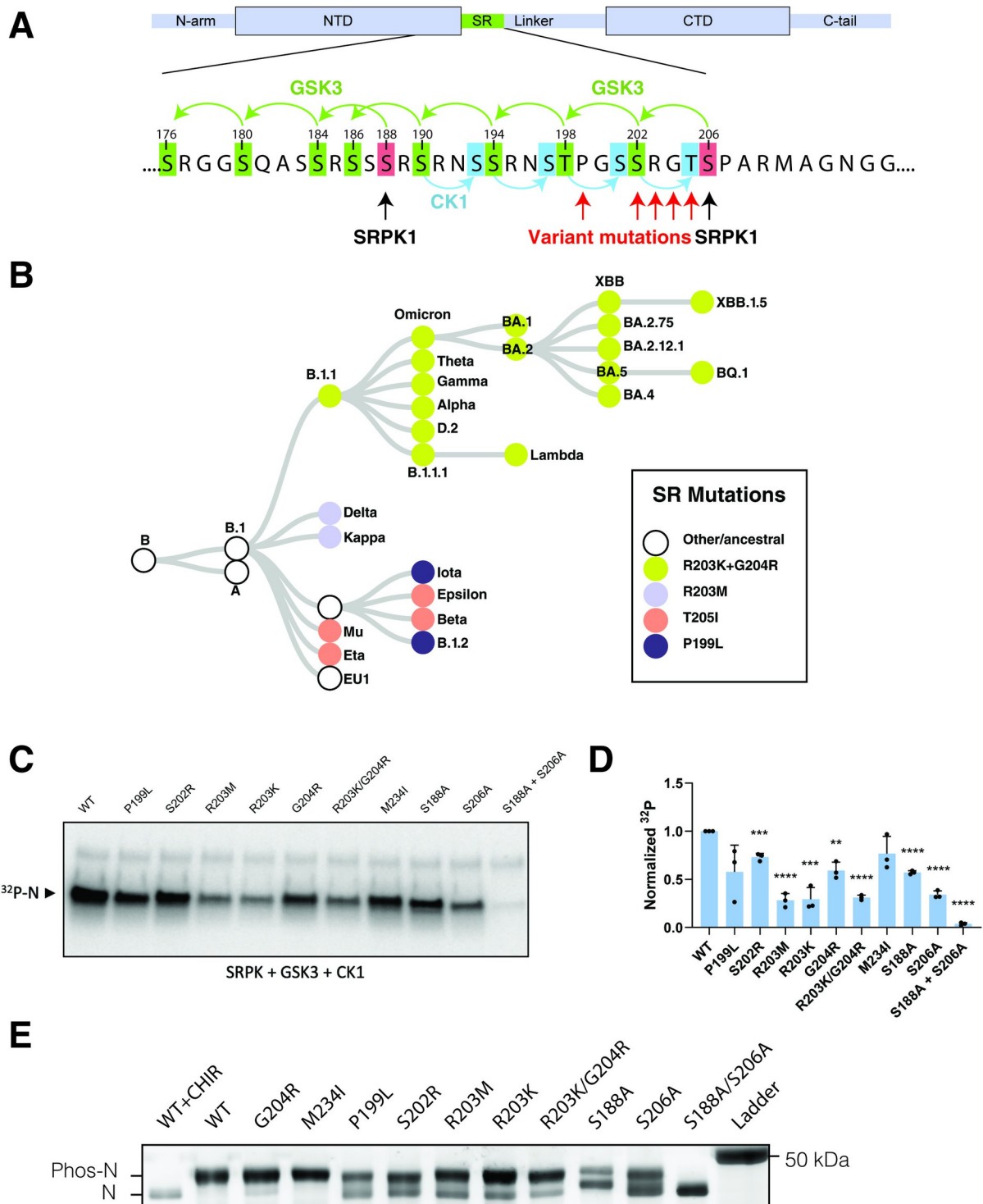

**Fig 1. SARS-CoV-2 variants have reduced N protein phosphorylation.** A) Map of SARS-CoV-2 N protein showing phosphorylation sites within the serine/arginine rich motif (SR). Proposed phosphorylation sites and kinases indicated with arrows. Location of common mutations found in circulating SARS-CoV-2 lineages indicated with red arrows (P199L, S202R, R203M, R203K, G204R, T205I). B) Schematic illustration of the phylogeny of circulating variants with predominant SR mutations indicated by color. C) Autoradiography ($^{32}$P) and quantification (D) of purified N variants after in vitro phosphorylation with SRPK, GSK3 and CK1 using $^{32}$P-γ-ATP (see S1A Fig). The blot is representative of 3 independent experiments and the quantification represents 3 independent experiments shown as mean +/- SD normalized to WT. *, p<0.05; **, p<0.01; ***, p<0.001; ****, p<0.0001 by Student's T-test compared to the WT control. E) SDS-PAGE gel of immunoprecipitated strep-tagged N-protein variants isolated from transfected 293T cells. Gel visualized by Krypton staining (see S1D Fig).

structural proteins and a reporter transcript encoding luciferase into producer cells (293T, Fig 2A). A 2-kb RNA packaging signal, termed T20, in the 3' UTR of the reporter transcript allowed specific packaging and delivery of the luciferase RNA in VLPs. We added filtered VLP-containing supernatants from producer cells to receiver cells (293T cells overexpressing ACE2 and TMPRSS2) to initiate luciferase expression (Fig 2A). We previously showed that this system recapitulates SARS-CoV-2 viral assembly as well as the impact of N protein mutations on assembly efficiency [1]. We selected a subset of N protein variants that have high (ancestral, G204R and M234I) or low (P199L, S202R and R203M) phosphorylation in human cells (Fig 1E) for VLP studies. We found that low-phosphorylation mutations enhanced viral assembly (P199L, S202R and R203M) and high-phosphorylation mutations decreased viral assembly (ancestral, G204R and M234I) (Fig 2B). Furthermore, after treatment with a specific GSK-3 inhibitor, SB216763 (10 μM), all six variants assembled efficiently and to a comparable level with a 10-fold enhancement of the low assembly variants and no significant enhancement of the high assembly variants (Fig 2B). We then comprehensively tested several GSK-3 inhibitors (SB216763, CHIR98014, BRD0703, BRD3731) at concentrations ranging from 20 nM to 20 μM with six N protein mutants. Three of four GSK-3 inhibitors caused similar enhancements in VLP assembly for the lower assembly variants (ancestral, G204R and M234I) while BRD3731 failed to have an effect (S2A–S2D Fig). BRD3731 has moderate selectivity for GSK3β over GSK3α suggesting that GSK3α may be more important for N protein phosphorylation [20]. The effects of inhibitors targeting SRPK1 (Alectinib, SPHINX31 and SRPKIN1) were less conclusive due to the high toxicity of these drugs although Alectinib and SPHINX31 did appear to narrow the difference between the high packaging and lower packaging N protein variants (S2E–S2G Fig). In contrast, inhibitors targeting other kinases, AKT (ipatasertib), or CK1 (TMX4117 and CK1-IN-1), had no specific effects (S2H–S2J Fig). We elected to only use CHIR98014 from this point on given its high potency in enhancing the assembly of lower assembly N proteins (ancestral, G204R and M234I) at concentrations that are not cytotoxic. We also tested whether GSK-3 inhibition would impact N protein from SARS-CoV-1, which has ~89% sequence identity with SARS-CoV-2 SR-rich domain [4]. We generated VLPs using S, M, E and the packaging signal from SARS-CoV-2 but substituted the N protein from SARS-CoV-1 and observed similar dose-dependent enhancement in VLP assembly for both SARS-CoV-2 and SARS-CoV-1 N proteins (Fig 2C). This suggests that phosphorylation could inhibit assembly of other coronaviruses as well.

To further validate the role of phosphorylation, we generated SARS-CoV-2 N protein variants with mutations targeting residues critical for GSK-3 phosphorylation. Individual alanine substitutions at S206, S202 and T198 resulted in enhancement in VLP-induced luminescence by 22.7-, 11.6- and 2.7-fold respectively (Fig 2D). Glutamic acid mutations that mimic the charge of a phosphoserine but would still disrupt further phosphorylation by GSK-3 had more moderate effects on VLP-induced luminescence (Fig 2D). On the other hand, the VLP luminescence signal was not significantly affected by alanine or glutamic acid substitutions at the adjacent sites of T205, S201 and S197 (Fig 2E). These results suggest that phosphorylation of N protein at S206, S202 and T198 inhibits assembly of SARS-CoV-2 VLPs by disrupting phosphorylation by GSK-3.

## N protein phosphorylation enhances viral replication

We then asked whether the reduced phosphorylation of N protein has deleterious effects at other stages of viral replication. The SR region is highly conserved in coronavirus N proteins [5] and previous studies suggest that blocking GSK-3 phosphorylation inhibits viral replication [5,21]. To test whether phosphorylation of N protein affects post-entry viral replication, we

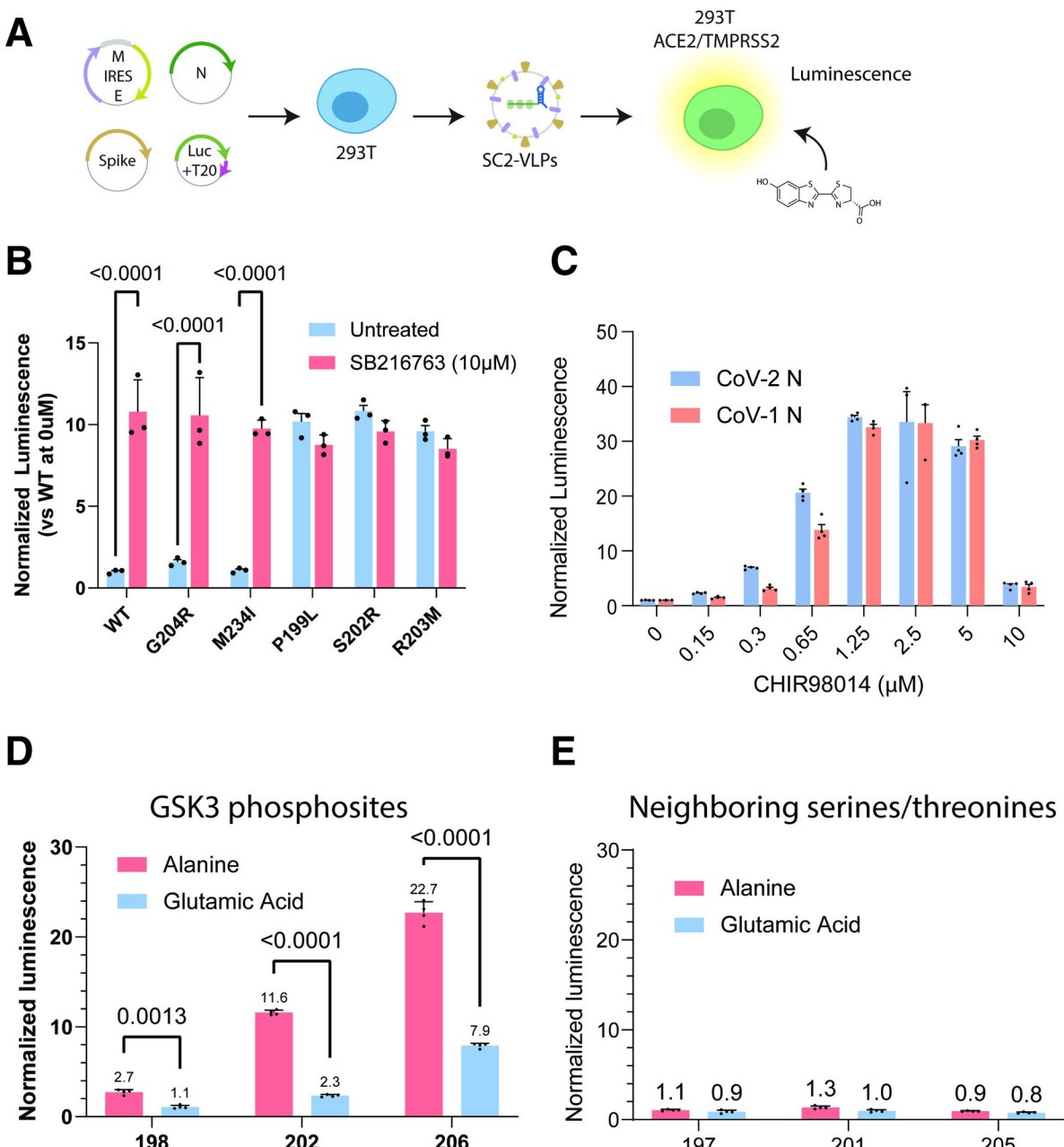

**Fig 2. N protein phosphorylation inhibits virus-like particle assembly.** A) Process flow for generating and testing SARS-CoV-2 virus-like particles packaging luciferase (VLPs). B) Normalized luminescence signal induced by VLPs generated using six variant N proteins with and without treatment with SB216763, a GSK3 inhibitor, at 10 μM. P199L, S202R and R203M have previously been shown to enhance VLP assembly [1] while G204R and M234I have similar packaging efficiency as the ancestral protein. The data are presented as mean +/- SD of three independent experiments. P values are indicated for piecewise comparisons using Student's T-test. C) VLP-induced signal generated using SARS-CoV-2 or SARS-CoV-1 N protein incubated with varying concentrations of CHIR 98014, another GSK3 inhibitor. The data are presented as mean +/- SD of four independent experiments. P values are indicated for piecewise comparisons using Student's T-test. D and E) Normalized luminescence signal from VLPs generated using N variants with alanine or glutamic acid substitutions at (D) critical GSK3 phosphorylation sites (198, 202, 206) or (E) neighboring sites (197, 201, 205). The data are presented as mean +/- SD of four independent experiments. P values are indicated for piecewise comparisons using Student's T-test.

developed a single cycle infection assay for SARS-CoV-2. We first generated a bacterial artificial chromosome (BAC) vector containing a replication-deficient SARS-CoV-2 genome (replicon) using previously described methods [22]. Prior studies have shown that deletion of S or N proteins can be used to generate SARS-CoV-2 replicons that maintain high levels of single cycle infection [23–25]. We substituted the N gene with a gene encoding NanoLuc luciferase and enhanced green fluorescent protein (Nluc-P2A-EGFP) so that the resulting genome would require N protein complementation to replicate (termed ΔN replicon, Fig 3A). The large size of the BAC vector limits the efficiency of transfection resulting in replication within less than 1% of cells. To improve the reproducibility of the assay, we co-transfected N protein along with the ΔN replicon for the first cycle in BHK21 cells to generate N protein-deleted single-cycle infectious particles (ΔN SCIPs) used for subsequent infections. These particles are able to infect cells with higher efficiency than observed with transfection but still require N protein complementation to assemble new particles and spread cell to cell (Figs 3A and S3A).

To assess the impact of N protein phosphorylation on post-entry replication within cells, we needed to introduce a variant of the N protein into the cells targeted for infection. However, expressing the full-length N protein in infected cells enables the formation of new particles and triggers multiple infection cycles that confound the results (S3A Fig). We first attempted to curb multiple cycles by adding bebtelovimab, a highly effective S-neutralizing antibody, after a 4-hour incubation with SCIPs. However, we observed clusters of GFP-expressing cells with or without bebtelovimab, indicating ongoing multi-cycle infection (S3A Fig). As an alternative approach, we examined the possibility of assembly-deficient N protein variants. Previous research by Kuo et al [26,27]. demonstrated that the C-tail of coronavirus N proteins binds to the membrane (M) protein and plays a crucial role in viral particle assembly. Importantly, swapping the C-tail domain of SARS-CoV with the corresponding sequence from a distantly related coronavirus (mouse hepatitis virus, MHV) did not impact viral replication, as long as the M proteins of the two viruses were also exchanged. This suggests that a domain swap of the N protein C-tail with MHV sequence should disrupt particle assembly and not compromise the role of N protein on viral RNA replication.

To validate this approach, we generated multiple domain swaps of SARS-CoV-2 N protein with homologous domains of the N protein from MHV (S3B Fig). In the VLP assay, domain swaps of the CTD (termed N-D4) and the C-tail (termed N-D5) abolished packaging validating these N proteins as assembly-deficient (S3C Fig). In cells treated with SCIPs, we found that cells were infected even in the absence of any N protein expression and that this infection was inhibited by nirmatrelvir (Pax) suggesting that N is not essential for post-entry replication for SARS-CoV-2 (Fig 3B). However, cells transfected with N-D5 produced higher levels of luminescence and brighter GFP signal compared to cells treated with a control plasmid not expressing N protein. Importantly, we did not observe clusters of infected cells for cells treated with N-D5 and the kinetics of infection were consistent with single-cycle infection (S3A Fig). We therefore used N-D5 and mutants within this construct for all subsequent single cycle infection experiments.

We used ΔN SCIPs to infect 293T-ACE2/TMPRSS2 cells expressing N-D5 or a vector control (CMVpA) for 4 hours and measured NanoLuc luminescence 24 hours after infection. Cells pre-treated with nirmatrelvir (Pax), a viral main protease (MPro) inhibitor, produced 89-fold lower luminescence than cells without Pax treatment suggesting that the luminescence signal represented genuine infection (Fig 3B). Cells transfected with N-S188A/S206A-D5, a phosphorylation defective mutant, or a vector control produced 5-fold lower luminescence compared to cells infected with N-D5 (Fig 3B). These results suggest that phosphorylation of N enhances post-entry replication. Most naturally occurring N protein mutants moderately reduced luminescence compared to ancestral N-D5 (Fig 3C), consistent with the fact that they

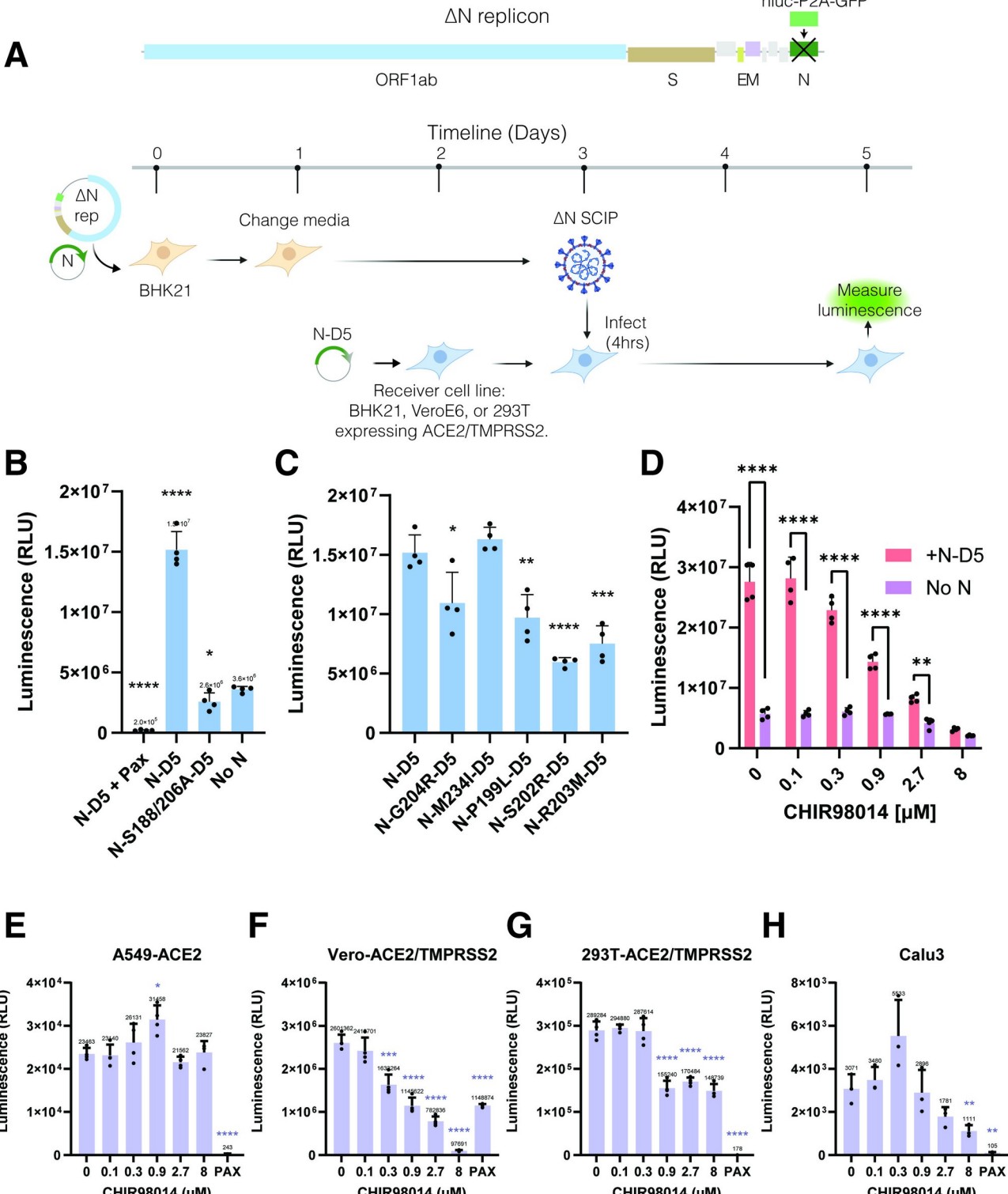

**Fig 3. N protein phosphorylation enhances single cycle infection.** A) Design of SARS-CoV-2 N-deleted replicon and process flow for single cycle infection. Clip art was created with BioRender.com with permission. B) and C) Luminescence from 293T-ACE2/TMPRSS2 cells transfected with packaging defective N mutants and infected with N-deleted single cycle infectious particles (ΔN-SCIPs). Pax indicates treatment with 3 μM nirmatrelvir. D) Luminescence from cells transfected with packaging defective ancestral N protein or a control plasmid and treated with varying concentrations of CHIR98014, a GSK3 inhibitor. E-H) Luminescence from cells infected with nanoluciferase expressing ΔS replicon encoding ancestral N sequence and

treated with varying concentrations of CHIR98014. All conditions with CHIR98014 or nirmatrelvir treatment were pre-treated 24 hours prior to infection, during infection and post-infection at the same concentration. Vero-ACE2/TMPRSS2 cells were additionally treated with 2μM CP-100356 (Millipore Sigma) to inhibit the P-glycoprotein efflux pump overexpressed in this cell line. For B-H, the data are presented as mean +/- SD of four (B-G) or three (H) independent experiments. *, $p < 0.05$; **, $p < 0.01$; ***, $p < 0.001$; ****, $p < 0.0001$ by Student's T-test.

partially inhibit phosphorylation (Fig 1E). In addition, GSK-3 inhibition by CHIR98014 reduced the luminescence signal in a dose-dependent manner for N-D5-expressing but not for mock transfected cells (Fig 3D).

To investigate the generalizability of our findings, we examined the effect of phosphorylation in other cell types and using a different infection system. We first tested the effect of N protein mutations in BHK-ACE2 and VeroE6-ACE2/TMPRSS2 cell lines using ΔN SCIPs and observed similar but less pronounced effects (S3D and S3E Fig). We also validated whether inhibiting phosphorylation with an otherwise unmodified N protein would have similar effects using ΔS SCIPs, as described previously [22], treated with CHIR98014 and once again observed reduced replication with increasing concentrations of CHIR98014 (Fig 3E–3H). Overall, these results suggest that single-cycle infection does not require N but is enhanced by the presence of phosphorylated N.

## N protein mutations (R203M, S202R, S206A) improve SARS-CoV-2 viral fitness

Given the opposing impacts of N protein phosphorylation on assembly and replication, we next examined how the modulation of phosphorylation affected overall viral fitness. We previously found that R203M and S202R substitutions in the Washington isolate 1 background (WA1) increased viral fitness more than 50-fold [1]. To test whether this enhancement was directly linked to phosphorylation, we generated infectious clones with NanoLuc luciferase substituted in place of Orf7a/b. We compared the replication of WA1 (high phosphorylation) and Delta variant (moderate phosphorylation) viruses in the presence of CHIR98014 in A549-ACE2 cells. Luminescence signal from WA1 increased in a dose-dependent manner with increasing CHIR98014 concentrations up to 5 μM while the Delta variant was unaffected (Fig 4A). These data are consistent with our observations of a large enhancement in VLP assembly (Fig 2C) and a modest decrease in SCIP replication (Fig 3D) for the ancestral N protein in the presence of CHIR98014. We next tested whether reduced phosphorylation would provide a fitness benefit in other cell lines. R203M and S202R substitutions in the WA1 background increased viral titers 72 hours after infection in A549-ACE2, Vero-TMPRSS2, and Huh7.5-ACE2/TMPRSS2 cell lines suggesting this enhancement is not restricted to our model cell lines (Fig 4B).

Small differences in viral growth rate may not be detectable by measuring viral titers in independent experiments so we additionally developed a competitive fitness assay to test the impact of substitutions that may have more moderate effects on viral fitness. We generated luciferase expressing viruses containing mutations R203M, S206A or M234I in the WA1 background and co-infected cells with these substitutions alongside the ancestral WA1 isolate at indicated MOI ratios (Fig 4C). We then quantified the relative abundance of these substitutions compared to WA1 using either Sanger sequencing (Fig 4D) or a qPCR assay with probes designed to detect single nucleotide substitutions (Fig 4E–4G). The R203M substitution outcompeted the ancestral sequence within the first two passages even when introduced at 100-fold lower abundance based on MOI than the ancestral virus (Fig 4D). We also tested viruses with R203M and S206A substitutions at equal MOI against WA1, with and without CHIR98014 and found that these clones outcompeted WA1 over three passages but not in the

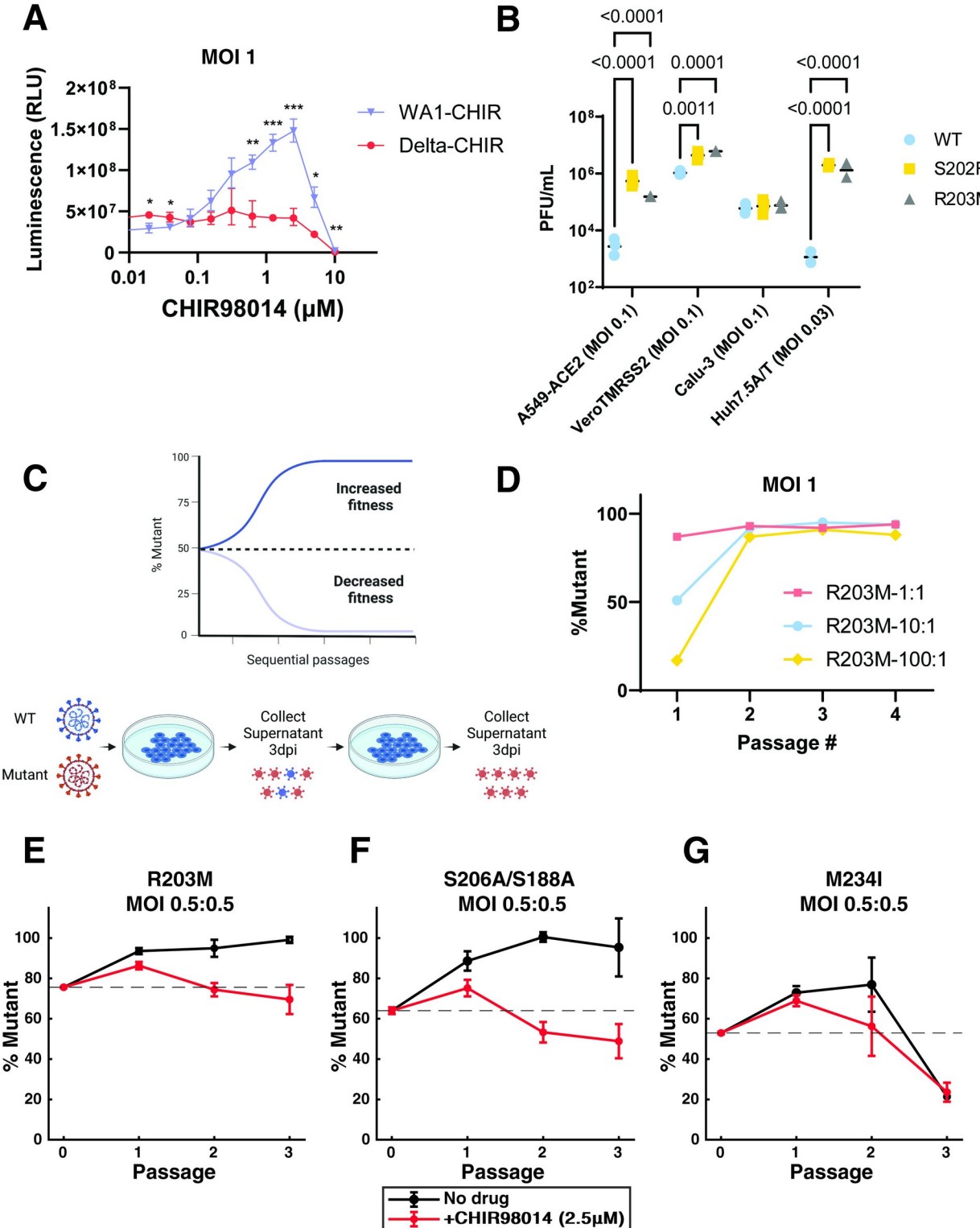

**Fig 4. N protein phosphorylation inhibits viral fitness.** A) Luminescence from cells infected with WA1 and Delta infectious clones with NanoLuc luciferase inserted in place of Orf7a and Orf7b. Cells were treated with varying concentrations of CHIR98014 as indicated. WA1 contains ancestral N protein while Delta contains the R203M substitution which reduces N protein phosphorylation. The data are presented as mean +/- SD of three independent experiments. *, p<0.05; **, p<0.01; ***, p<0.001 for piecewise comparisons conducted by Student's T-test. B) Plaque forming units of infectious clones encoding ancestral (WT), S202R or R203M mutations within the N protein, 72 hours post infection in

A549-ACE2, Vero TMPRSS2, Calu-3 and Huh7.5 ACE2/TMPRSS2 cells. The data are presented as mean +/- SD of three (A549-ACE2, Vero TMPRSS2, and Calu-3) or two (Huh7.5 ACE2/TMPRSS2) independent experiments conducted in triplicate. P values from Student's T-test are indicated in the panel. C) Schematic of viral competition assay. Clip art was created with BioRender.com with permission. D) Quantification of a pilot competition experiment between R203M mutant virus on WA1 background using Sanger sequencing after co-infection with 1:1, 10:1, or 100:1 ratios of WA1:R203M on A549-ACE2 cells. The data is presented as % of mutant based on Sanger sequencing from a single biological experiment. E-G) Quantification of competition between mutant viruses (R203M, S188A/S206A and M234I) on WA1 background relative to the WA1 quantified through qPCR. Gray lines indicate results of competition in untreated cells while red lines indicate treatment with CHIR98014 at 2.5 μM. The data are presented as mean +/- SD of three independent experiments.

presence of CHIR98014 (Fig 4E and 4F). Note that since we measured the abundance of substitutions by qPCR, this ratio was not 50% at passage 0 since RNA abundance is not always equal to MOI. The M234I infectious clone on the other hand was not able to outcompete the ancestral variant under either condition, consistent with our observations that this mutation does not affect phosphorylation or assembly (Fig 4G). These data suggest that the reduced phosphorylation in mutants R203M and S202R provides an overall fitness advantage and that the improvements in assembly outweigh the deficits in replication for these mutants.

## Natural N protein truncation enhances both viral replication and viral particle assembly

A common outcome of adaptive conflict between two functions of a protein is the evolution of a duplicate protein with a specialized role. Intriguingly, the B.1.1 lineage which includes the Alpha, Gamma, Omicron and several other variants has been shown to express a shortened N protein termed N* [28–31]. In these variants, a mutation at nucleotides 28881–28883: ggg>aac generates amino acid substitutions R203K/G204R and creates a new transcriptional regulatory sequence (TRS) that leads to the expression of an additional viral RNA transcript (Fig 5A). N* is translated starting at methionine 210 of the N open reading frame and lacks the entire SR region and associated phosphorylation sites. Recent studies suggest that this truncated protein is expressed during infection although at low abundance [30,31]. Since N* would lack phosphorylation but maintain the CTD and C-tail regions known to be critical for packaging, there is a possibility that it could allow viral assembly even in the presence of high levels of phosphorylation in the full-length N protein required for efficient replication.

To test whether N* can package RNA and resolve adaptive conflict in N packaging, we tested its assembly by generating VLPs using N* as well as a range of other truncated N-variants (Fig 5B and 5C). VLPs generated with N* induced luciferase expression more efficiently than the ancestral N protein. In contrast, all other truncations tested disrupted assembly including N-DE (aa258-419) and N-CD (aa 175–362) implying that intrinsically disordered regions on both sides of the CTD are required for packaging (Fig 5B and 5C). We also concentrated VLPs produced from cells co-expressing N* and full-length N and found that N* was selectively incorporated into VLPs relative to full-length N (Figs 5D and S4). We next compared the effect of N* or GSK-3 inhibition on VLP assembly. We generated VLPs with varying ratios of full-length N and N* and tested each condition with and without CHIR98014 (1.25μM). We found that N* enhanced assembly in all cases but with lower potency than full-length N along with CHIR98014 (Fig 5E). These results imply that N* is capable of assembly that is not affected by GSK-3 phosphorylation, but is not as efficient as unphosphorylated full-length N implying the possibility of further evolution for this purpose.

To validate these findings, we also generated an infectious clone that encodes the N* transcript but does not modify phosphorylation in the SR region. By mutating the ancestral codons AGG-GGA-ACU to AG<u>A-CGA-AC</u>U, we generated a clone that encodes a new TRS site

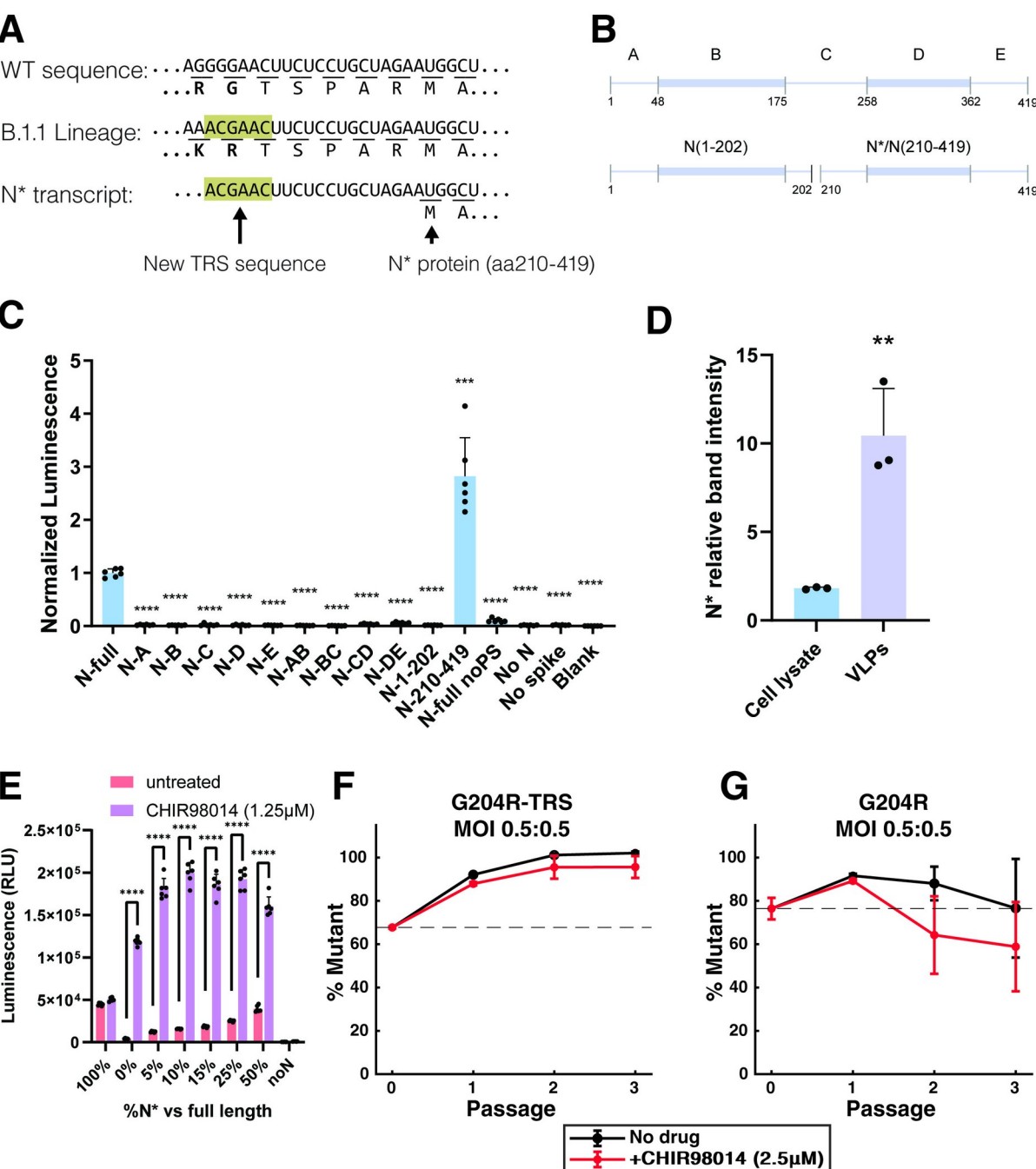

**Fig 5. N* improves viral fitness independent of phosphorylation.** A) Annotation of the 28881–28893: ggg>aac mutation showing the generation of the "ACGAAC" TRS sequence. B) Schematic of truncated N proteins produced to test effects on assembly. C) Luminescence from VLP treated 293T-ACE2/TMPRSS2 cells. VLPs were generated with indicated truncated N proteins. Only N* (N:210–419) is competent for delivering RNA. The data are presented as mean +/- SD of one independent experiment conducted in six replicates. ***, p<0.001; ****, p<0.0001 by Student's T-test compared to wildtype N. D) Quantification of Western blot (streptactin-HRP) from producer cell lysate and purified VLPs showing relative N* abundance compared to full length N protein. Both proteins contain a C-terminal 2xStrep-Tag. The data are presented as mean +/- SD of one independent experiment conducted in triplicate and the blot is shown in S4 Fig. **, p<0.01 by Student's T-test. E) Luminescence from VLP treated cells with VLPs generated with combinations of N* and full length N protein and with or without treatment with CHIR98014. The data are presented as mean +/- SD of one independent experiment conducted in six replicates. P values are indicated for piecewise comparisons conducted with Student's T-test. F, G) Viral competition assays of G204R-TRS virus (expresses N*) and G204R virus (no N*) with WA1 infectious clone with or without CHIR98014 treatment as measured by qPCR. The data are presented as mean +/- SD of three independent experiments.

(underlined) along with a G204R substitution which has only modest effects on phosphorylation (Fig 1E). Termed G204R-TRS, this clone outcompetes the WA1 isolate with or without GSK-3 inhibition (Fig 5F and 5G). In comparison, an infectious clone with G204R without the TRS failed to outcompete the WA1 isolate under either condition. These data show that N* has a functional role in enhancing viral assembly and reducing its dependence on N protein phosphorylation.

## Discussion

In this study, we investigated the impact of N protein phosphorylation on SARS-CoV-2 assembly and replication, and how naturally occurring mutations affect this process. A large number of mutations occur within a region of N protein that undergoes high levels of phosphorylation, hinting at a shared mechanism. Multiple lines of evidence support the conclusion that mutations within this region, including S202R, R203M and R203K, significantly reduce N protein phosphorylation. We tested these and other naturally occurring mutations first in a biochemical assay and found that they decrease phosphorylation when treated with a combination of GSK3, SRPK1 and CK1. These mutations also reduced N protein phosphorylation in transfected cells and enhanced the assembly of VLPs as effectively as GSK3 inhibitors. All of these observations point to the role of these mutations in reducing phosphorylation and enhancing VLP assembly. These findings are also consistent with prior literature suggesting that phosphorylation of N protein limits its assembly into particles [8,10–12].

Notably, the SR region with its distinct phosphorylation motif is highly conserved in all known coronaviruses suggesting an important role in viral fitness. Indeed, the SR region is highly conserved in SARS-CoV-1 and related viruses as well, and our results suggest that inhibition of phosphorylation could enhance assembly in these viruses as well. However, this raises the question of whether there are other beneficial effects of N protein phosphorylation that justify the fitness cost of retaining this region in all known coronaviruses. To test whether N protein phosphorylation would be beneficial in the remaining steps of the viral lifecycle, we developed a single-cycle infection assay that isolated post-entry replication. We found that single cycle infection does not require N but is moderately enhanced by phosphorylated N. Our assay for measuring the effect of N on post-entry replication may underestimate this contribution since not all cells are transfected with N prior to infection with single cycle infectious particles. This observation is also consistent with previous research studies suggesting that targeting GSK3 phosphorylation leads to lower levels of replication for SARS-CoV-2 [4,5,21]. Our results point to an adaptive conflict between assembly and replication, where phosphorylation of N protein enhances replication at the cost of impaired assembly.

One strategy for resolving a conflict between two functions of a protein is gene duplication followed by specialization. Intriguingly, the B.1.1 lineage, which includes most variants of concern, expresses an additional truncated N protein (termed N*) that lacks phosphorylation sites yet retains regions critical for packaging [11,28]. Our results suggest that N* can efficiently package RNA despite high endogenous GSK3 activity, resolving the adaptive conflict between N protein phosphorylation and assembly. Although N* has lower packaging efficiency compared to unphosphorylated full-length N, the combination of N* and full-length N provides a fitness benefit independent of cellular GSK3 activity. Our findings suggest that N* could be a step towards an evolutionary bifurcation where the functions of N are split between two distinct proteins leading to variants that are more efficient in both packaging and replication. However, it is not clear whether N* is a major driver of enhanced assembly for currently circulating variants since previous work suggests that it is expressed at a low level [30,31] and because it co-occurs with mutations that also reduce phosphorylation.

Our results also suggest that GSK-3 inhibitors may not be an ideal antiviral therapy for SARS-CoV-2 as proposed in a few recent papers [5,21]. GSK-3 inhibitors have previously been developed for treatment of bipolar disorder, Alzheimer's and other neurodegenerative diseases with varying degrees of clinical use [32–34]. In the context of SARS-CoV-2, reducing GSK-3 activity inhibits viral genome replication which could potentially limit viral pathogenesis. However, our results show that a reduction in GSK-3 activity could also improve viral fitness by enhancing viral assembly. Circulating variants vary in the extent of N protein phosphorylation and this phosphorylation level may change in future variants. Our results suggest that GSK-3 inhibitors would be proviral for treating variants with high levels of N protein phosphorylation but could potentially work against variants with low levels of phosphorylation. Current circulating variants have moderate levels of phosphorylation and a truncated N* protein that enhances assembly regardless of phosphorylation. These variants are likely to be unaffected by moderate changes in GSK-3 activity, suggesting this approach is unlikely to work against current SARS-CoV-2 variants.

Contrary to our results, two recent studies have reported that R203K/G204R [3] and S202R [35] substitutions lead to increased N protein phosphorylation. Both studies used PhosTag gels to evaluate phosphorylation, but this approach produced broad, difficult to interpret smears in our experiments due to the high degree of phosphorylation in N protein (S1C Fig). When we use an orthogonal biochemical phosphorylation assay, we find that the S202R mutation slightly reduces phosphorylation while the R203K/G204R mutation significantly reduces phosphorylation due mostly to R203K (Fig 1C). Furthermore, the S202R mutation in particular is unlikely to increase phosphorylation since it should block GSK3 phosphorylation at sites 202, 198, 194, 190 and 186 [4,5]. Despite this discrepancy, both studies primarily discussed the impact of these mutations on viral fitness and observed increased fitness in both cases similar to our findings for S202R (Fig 4B and [1]) and R203K/G204R [36].

In conclusion, our findings provide insights into the opposing roles of GSK-3-mediated N protein phosphorylation in SARS-CoV-2 assembly and replication. In addition to receptor usage, entry efficiency, and degree of immune evasiveness, N protein phosphorylation may have important implications for understanding the molecular mechanisms underlying viral evolution and fitness, and may inform the development of antiviral therapies that target the delicate balance between assembly and replication in SARS-CoV-2 and other coronaviruses. Lastly, given the conservation of the GSK-3 phosphorylation site in other sarbecoviruses, a future pandemic coronavirus could undergo a similar pattern of evolution. Our results suggest that monitoring of mutations that modulate N protein phosphorylation state in pre-pandemic SARS-related viruses may provide insight about human adaptation of these viruses.

## Materials and methods

| REAGENT or RESOURCE | SOURCE | IDENTIFIER |
|---|---|---|
| Antibodies | | |
| Strep-Tactin HRP conjugate | IBA lifesciences | Cat#2-1502-001 |
| Bebtelovimab Biosimilar–Anti-SARS-CoV-2 Spike Protein mAb–Research Grade | ProteoGenix | Cat#PX-TA1750-100UG |
| Chemicals, peptides, and recombinant proteins | | |
| Krypton Fluorescent Protein Stain | Thermo Scientific | Cat#46630 |
| CHIR-98014 | MedChemExpress | Cat#HY-13076 |
| BRD3731 | MedChemExpress | Cat#HY-124607B |

(*Continued*)

| | | |
|---|---|---|
| BRD0705 | MedChemExpress | Cat#HY-116830 |
| SB-216763 | MedChemExpress | Cat#HY-12012 |
| SPHINX31 | MedChemExpress | Cat#HY-117661 |
| Ro-3306 | MedChemExpress | Cat#HY-12529 |
| Ipatasertib | MedChemExpress | Cat#HY-15186 |
| Nirmatrelvir | Selleckchem | Cat#S9866 |
| Phos-tag (TM) Acrylamide | FUJIFILM wako chemicals | Cat#AAL-107 |
| Transit-x2–1.5 mL | Mirus Bio LLC | Cat#MIR6000 |
| Passive Lysis Buffer | Promega | Cat#E1941 |
| SRPK1 Kinase Enzyme | Promega | Cat#VA7558 |
| GSK3β Kinase Enzyme | Promega | Cat#V1991 |
| CK1ε Kinase Enzyme | Promega | Cat#V4160 |
| Hoechst 34580 (5mg) | Life Technologies | Cat#H21486 |
| Polyethylenimine, Linear, MW 25000, Transfection Grade | Polysciences | Cat#23966–1 |
| Critical commercial assays | | |
| MagStrep "type3" XT magnetic beads | IBA lifesciences | Cat#2-4090-002 |
| Illumination Lyophilized Firefly Luciferase Enhanced Assay Kit | Gold Biotechnology | Cat#I-935-1000 |
| CellTiter-Glo 2.0 Cell Viability Assay | Promega | Cat#G9242 |
| Nano-Glo Luciferase Assay System | Promega | Cat#N1120 |
| SUPERSEP 50UM7.5% 17WELL83X100 | Wako Chemical | Cat#198–17981 |
| Luna Probe One-Step RT-qPCR 4X Mix with UDG | New England Biolabs | Cat#M3019X |
| Experimental models: Cell lines | | |
| 293T-ACE2/TMPRSS2 | This paper | N/A |
| A549-ACE2 | This paper | N/A |
| Oligonucleotides | | |
| Primer qPCR fwd: CAGTCAAGCCTCTTCTCGTTC | This paper | N/A |
| Primer qPCR rev: CCTTGTTGTTGTTGGCCTTTAC | This paper | N/A |
| Primer Probe1-188: 5'-SUN-CG+AC+T+ACG+TG+A +TG-IABkFQ-3' | This paper | N/A |
| Primer Probe2-204: 5'-FAM-AG+A+AG+T+TCCCC+T +ACT-IABkFQ-3' | This paper | N/A |
| Primer Probe3-234: 5'-Cy5-GCA+A+A+A+TG+TC +TGGTA-IAbRQSp-3' | This paper | N/A |
| Recombinant DNA | | |
| N Rep | This paper | N/A |
| N-D5 | This paper | N/A |
| N-CoV-1 | This paper | N/A |
| Luc-T20 | Syed et al.[1] | Addgene plasmid#177941 |
| CoV2-M-IRES-E | Syed et al.[1] | Addgene plasmid#177938 |
| CoV2-N-WT-Hu1 | Syed et al.[1] | Addgene plasmid#177937 |
| CoV2-Spike-EF1a-D614G-N501Y | Syed et al.[1] | Addgene plasmid#177939 |

## Methods

Single Cycle Infectious Particle Production: BHK21 cells were seeded in 15 cm plates containing 20 mL of Dulbecco's modified Eagle's medium (DMEM) containing fetal bovine serum (FBS, 10%) and penicillin/streptomycin (1%) and incubated for 24 hours before transfection. 22.5 ug of plasmids N Rep (0.6), N-R203M (0.2), and VSV-G (0.2) at indicated mass ratios were diluted in 1.125 mL room temperature Opti-MEM. 67.5 ug of room temperature MIRUS TransIT-X2 was diluted in 1.125 mL Opti-MEM and used to resuspend plasmid dilutions. The diluted plasmid transfection mix was incubated at room temperature for 20 minutes and added dropwise onto the cells. Media was refreshed after 24 hours. 72 hours after transfection, the supernatant was removed and filtered with a 0.45 μm pore size polyethersulfone syringe filter. SCIP-containing supernatant was used immediately after collection or frozen in liquid nitrogen and stored at -80°C. Note: multiple freeze thaw cycles using liquid nitrogen had minimal effect on titer.

## Single cycle infectious particle infection

BHK21-ACE2 cells were seeded at 15,000 cells/well or 293T-ACE2/TMPRSS2 cells were seeded at 25,000 cells/well in poly-L-lysine coated 96 well plates in 150 μL of Dulbecco's modified Eagle's medium (DMEM) containing fetal bovine serum (FBS), penicillin/streptomycin, and 10 μg/mL blasticidin. 24 hours after seeding, cells were transfected with 0.15 μg of N plasmid in each well using 0.45 μL of Mirus TransIT-X2 transfection reagent. In conditions requiring CHIR98014 or Nirmatrelvir, corresponding drug was added immediately after transfection. Media was removed 24 hours later and 100 μl SCIP-containing supernatant was added. Media was changed within 8 hours after infection. For luciferase measurement, 24 hours after media change, plates were gently shaken to mix the luciferase in the supernatant by hand. 50μl of supernatant from each well was transferred to an opaque white 96 well plate. Using a Tecan Spark, 30μL of NanoGlo lytic luciferase assay buffer was dispensed into each well and mixed for 60 seconds by shaking. 3 minutes after shaking, luminescence was measured using the auto attenuation function and 1000 ms integration time.

## Live cell imaging of SCIP infection

After the SCIP infection step, media was changed to imaging media (FluoroBrite DMEM + 10% fetal bovine serum + 1% penicillin/streptomycin + 0.5 μg/mL Hoechst 33458). SCIP infected cells were imaged every hour using an ImageXpress Micro confocal microscope on both Hoechst and GFP channels.

## Virus-like particles assay

293T cells were plated at 50,000 cells/well in 150μL of DMEM containing FBS and penicillin/streptomycin in 96 well plates and incubated for 24 hours. 1.5 μg of plasmids N (0.33), T20 (0.5), Sv4 (0.0033), and MIEv3 (0.165) at indicated mass ratios were diluted in 75 μL room temperature Opti-MEM. 4.5 μL of 1 mg/mL polyethylenimine (pH 7.0) was diluted in 75 μL Opti-MEM and used to resuspend plasmid dilutions. The diluted plasmid transfection mix was incubated at room temperature for 20 minutes. For transfection into 96 well plates, 18.75 μL of resulting transfection mix was added to each well and mixed by pipetting up and down carefully. 2 days after transfection, 65μL of supernatant was filtered using Pall 0.45 μm AcroPrep Advance Plate, Short Tip filter plates (polyethersulfone) by centrifuging the filter plate at 100g for 2 minutes. 50,000 293T-ACE2/TMPRSS2 cells were added to the VLP-containing filtrate and incubated for 24 hours. The supernatant was removed and 20 μL of

Promega passive lysis buffer was added to the cells. The cells were then incubated on a shaker for 20 minutes. Using a Tecan Spark, 30 μL of luciferase assay buffer was dispensed into each well and mixed for 60 seconds by shaking. Luminescence was measured using the auto attenuation function and 1000ms integration time.

## Strep-tag Immunoprecipitation

MagStrep type3 XT beads were washed 3 times with Strep-Tactin XT wash buffer and once with Na-TBST buffer (1M NaCl, 8 mM Tris, 0.08% Tween20, pH7.6). 25 μL of 5M NaCl was added to 100 μL of lysates containing strep-tagged N and vortexed to increase stability of N protein assemblies. To these lysates, 10 μL of 10% of washed MagStrep type3 XT bead solution was added and mixed thoroughly by vortexing. Samples were then incubated at 4°C on a rotating rack for 30 minutes. Sample tubes were then placed in a magnetic rack and unbound lysate was pipetted off. The beads were washed three times with Na-TBST buffer and once with Strep-Tactin XT wash buffer. Buffer was removed and 100 μL of 1X Laemmli buffer was added to the beads. The sample tubes were vortexed thoroughly and incubated at 95°C for 10 minutes to release bound N protein. The sample tubes were then placed in a magnetic rack and the sample containing supernatant was removed.

## SDS-PAGE Gel staining with Krypton

After lysates were run on a 7.5% acrylamide gel, the gel was rinsed with transfer buffer (25mM Tris, 192 mM glycine, 10% methanol) containing 1X EDTA for 5 minutes. Two different staining procedures were used. For the Krypton treatment, gels were washed in gel fixing solution (40% ethanol and 10% acetic acid in ultrapure water) on a rocker for 5 minutes twice. The gels were then rinsed with ultra-pure water. The gels were then immersed in 1X Krypton protein stain and placed on a rocker for 15 minutes and rinsed with ultrapure water. For the chloroform treatment, gels were rinsed with ultrapure water and then incubated in ultrapure water containing 0.5% chloroform for 5 minutes. The gels were imaged using a ChemiDoc MP Imaging System.

## N Protein bacterial expression

Ancestral and mutant N proteins were produced as described previously [9,10]. Briefly, a codon-optimized synthetic DNA (Integrated DNA Technologies, IDT) was inserted into a pET28 expression vector by Gibson assembly, fused to DNA encoding an N-terminal 6xHis-SUMO tag. Mutant N proteins were generated by site-directed mutagenesis. N proteins were expressed in E. coli BL21 Star (Thermo #C601003), grown in TB–Kanamycin to absorbance 0.6, and induced with 0.4 mM IPTG. Cells were harvested, washed with PBS, snap frozen in LN2, and stored at −80°C until use. Thawed cells were resuspended in buffer A (50 mM Hepes pH 7.5, 500 mM NaCl, 10% glycerol, and 6 M urea) and lysed by sonication. The lysate was clarified by centrifugation and bound to Ni-NTA agarose beads (QIAGEN #30230) for 45 min at 4°C. Ni-NTA beads were washed three times with ten bed volumes of buffer A and eluted with buffer B (50 mM Hepes pH 7.5, 500 mM NaCl, 10% glycerol, 250 mM imidazole, and 6 M urea). The eluate was concentrated in centrifugal concentrators (Millipore Sigma #UFC803024), transferred to dialysis tubing (Spectrum Labs #132676), and renatured overnight by dialysis in buffer C (50 mM Hepes pH 7.5, 500 mM NaCl, 10% glycerol). Recombinant Ulp1 catalytic domain (purified separately from E. coli) was added to renatured protein to cleave the 6xHis-SUMO tag, and cleaved protein was injected onto a Superdex 200 10/300 size-exclusion column equilibrated in buffer C. Peak fractions were pooled, concentrated, frozen in LN2, and stored at −80°C.

## Protein kinase reactions

Protein kinases were purchased from Promega (SRPK1: #VA7558, GSK-3β: #V1991, CK1ε: V4160). N protein (1.25 μM) was incubated with 80 nM kinase for 30 min at 30˚C in kinase reaction buffer (25 mM Hepes pH 7.5, 35 mM KCl, 10 mM MgCl2, 1 mM DTT, 0.5 mM ATP, and 0.001 mCi/ml 32P-γ-ATP). Reactions were quenched upon addition of SDS loading buffer for analysis by SDS-PAGE and autoradiography.

## Infectious clone competition assay

A549-ACE2 cells were infected with a 50:50 mixture of WA1 infectious clone in a biosafety level 3 lab and a mutant infectious clone (on WA1 background) with a final multiplicity of infection of 1 for 24 hours. 72 hours after infection, supernatants were passaged 1:10 on to new A549-ACE2 cells with the remaining supernatant saved for analysis. Each mutant was passaged three times in total and P0 supernatant (used for original infection) was used as a control. All supernatants were diluted 1:3 in trizol in deepwell 96 well plates to neutralize infectious particles and preserve RNA for further analysis.

## RNA extraction

RNA Extraction was done using Zymo's Direct-zol -96 RNA kit. Samples stored at -80˚C were thawed at room temperature in water baths for 30 minutes. In a chemical fume hood after trizol treated samples were thawed and vortexed gently. In empty deep well plates, 100 μL of 100% ethanol was added to each well. 100 μL of sample was transferred into the plate containing 100 μL ethanol per well. The Zymo-Spin I-96 plates were placed on collection plates. 200 μL of the sample-ethanol mixture was transferred into the spin plates and centrifuged at 2,500 g for 5 minutes. Flow-through was discarded. 400 μL of Direct-zol RNA PreWash was added to each well. Plates were spun down at 2,500 g for 5 minutes. Flow-through was discarded. This washing step was repeated. 800 μL of RNA Wash Buffer was added to each well. Plates were spun down at 2,500 g for 5 minutes. The Zymo-Spin I-96 plates were placed on elution plates and 25 μL of DNase/RNase-Free Water was added to every well. Plates were spun down at 2,500 g for 5 minutes to collect eluted RNA. Plates containing RNA were then sealed and stored at -80˚C until further analysis.

## Competition assay analysis by qPCR

qPCR assay was designed to detect relative abundance of ancestral and mutant sequences. Common forward and reverse primers were designed flanking the region containing all relevant mutants and three locked nucleic acid probes were designed to bind to the ancestral sequence centered on amino acids 188, 203 or 234 since all N mutations examined occur at these locations. For each 5 μL reaction, we used 0.5 μL extracted RNA, forward and reverse primers at 0.9 μM, Probe1-188 (Sun) and Probe2-204 (FAM) at 0.15 μM and Probe3-234 (Cy5) at 0.45 μM. We used NEB Luna Probe One-Step RT-qPCR 4X Mix with UDG (M3019X) with 15 minute hold at 50˚C, 30 second hold at 95˚C followed by 50 cycles of 95˚C for 5 seconds and 62˚C for 5 seconds on a QuantStudio 5 instrument. Positive and negative controls were plasmids encoding wildtype and each mutant sequence and the mutant fraction was calculated by comparing relative slopes of corresponding probes.

## Cloning for replicon and infectious clone plasmids

SARS-CoV-2 genome was cloned into pBAC plasmids using the pGLUE protocol as reported by Taha *et al.*[22] Briefly, the genome was split into 10 high copy fragment plasmids with all N

mutations made in fragment 10 by PCR and NEB HiFi Assembly protocol. N replicon was constructed by replacing the N gene in Fragment 10 with NanoLuc-P2A-eGFP preserving the original transcriptional regulatory sequence and flanked with the same BsaI recognition and overhang sequences. All fragment plasmids were verified by whole plasmid sequencing. Fragments 1–10 were then amplified from their respective plasmids by PCR and assembled along with the pBAC vector using Golden Gate assembly. The assembly was performed using 30 cycles of 37˚C for 5 min, followed by 16˚C for 5 min. The reaction was then incubated at 37˚C for 5 min and 60˚C for 5 min. 1 μL of the reaction was electroporated into EPI300 cells and plated onto LB + chloramphenicol plates and grown at 37˚C for 24 h. Colonies were picked and cultured in LB30 medium + 12.5 μg/mL of chloramphenicol for 12 h at 37˚C. 1 mL of the culture was diluted to 100 mL of LB30 medium + 12.5 μg/mL of chloramphenicol for 3–4 hours. The culture was diluted again to 400 mL of LB30 medium + 12.5 μg/mL of chloramphenicol + 1x Arabinose induction solution (Lucigen) overnight. pBAC infectious clone and replicon plasmids were extracted and purified using Zymo MaxiPrep kit and verified by whole plasmid sequencing (Plasmidsaurus or Priomordium labs).

## Infectious clone launch

250,000 BHK21 cells per well were seeded in 6-well plates. The next day, cells were transfected with 1.125 μg infectious clone plasmids and 1.125 μg N-R203M plasmid (addgene: 177952) of DNA and 7.5 μL of the transfection agent Mirus-TransIT-X2. Media was changed the following day and cells were incubated until cytopathic effect was observed. Virus-containing supernatants were collected and frozen. Infectious titer was determined by plaque assay on VeroE6-TMPRSS2 cells.

## Supporting information

**S1 Fig. N protein phosphorylation visualized by SDS-PAGE.** A) Coomassie blue stained gel image for the autoradiography image shown in Fig 1C. B) N protein variants affinity-purified from transfected 293T lysates visualized by Krypton staining on a 7.5% SDS-PAGE gel. CHIR indicates lysate purified from cells treated with 2.5 μM CHIR98014, a GSK-3 inhibitor. C) Same N protein variants visualized by Krypton staining on a PhosTag phosphate affinity gel. Note: molecular weight for ladder bands is not labeled since migration in PhosTag gels does not correspond directly with molecular weight. D) Full gel image for the gel used in Fig 1E showing several naturally occurring and engineered N protein variants isolated from 293T lysates. The last two lanes are N proteins isolated from E. coli with and without *in vitro* phosphorylation using GSK-3, CK1 and SRPK.
(TIF)

**S2 Fig. Effect of kinase inhibitors on VLP assembly with multiple N protein variants.** A-D) Left axis shows luminescence from cells treated with VLPs generated with N mutants: P199L, S202R, R203M, G204R, M234I and WT. Producer cells were treated with indicated concentrations (20 nM to 20 μM) of GSK-3 inhibitors: SB216763, CHIR98014, BRD0703, BRD3731. Right axis and black lines shows viability as measured by CellTiter-Glo luminescence from cells treated with the same concentrations of corresponding drugs but not with VLPs. E-H) Left axis shows normalized luminescence from cells treated with VLPs generated with N mutants: S188A, S188A/S206A, P199L, S202R, R203M, G204R, M234I and WT. Producer cells were treated with indicated concentrations (20 nM to 20 μM) of inhibitors targeting multiple kinases: Alectinib (SRPK1), SPHINX31 (SRPK1), SRPKIN1 (SRPK1), Ipatasertib (AKT), TMX4117 (CK1), CK1-in-1 (CK1). Right axis and black lines shows viability as measured by

CellTiter-Glo luminescence from cells treated with the same concentrations of corresponding drugs but not with VLPs. No replicates were used for the data in these panels.
(TIF)

**S3 Fig. Single cycle infection assay development and validation.** A) Live microscopy snapshots with nuclei shown in blue (Hoechst 34580) and GFP in green (eGFP, produced from infection) from SCIP infected cells (293T-ACE2/TMPRSS2). Cells were transfected with indicated N protein variant. Large green spots show infection of a cluster neighboring cells resulting from cell to cell spread. B) Schematic of domain swap variants of N proteins from SARS-CoV-2 and MHV-A59. Numbers indicate positions on SARS-CoV-2. C) Luminescence signal from VLP assay using domain swap N proteins. N-D5 has negligible signal indicating that it is packaging defective. D, E) Luminescence signal from SCIP infected BHK21-ACE2 (D) and VeroE6-ACE2/TMPRSS2 (E) that were transfected with N proteins with indicated mutations or treated with CHIR98014 at 3 μM. VeroE6-ACE2/TMPRSS2 were also treated with 2 μM of the P-glycoprotein efflux inhibitor CP-100356 due to their high endogenous efflux activity. The data are presented as mean +/- SD of one independent experiment conducted in three (C) and four (D and E) replicates. *, $p<0.05$; **, $p<0.01$; ***, $p<0.001$; ****, $p<0.0001$ by Student's T-test. F) Cytotoxicity data of CHIR98014 in indicated cells corresponding to treatments in Fig 3E–3H. Data is presented as mean +/- SD of three independent replicates conducted in one technical replicate. *, $p<0.05$ by Student's T-test compared to no treatment control.
(TIF)

**S4 Fig. Western blot of producer cell lysate and purified VLPs from N\* and N transfected cells.** N\* and N plasmids both contain C-terminal 2xStrepTag2 and are imaged using StreptactinXT-HRP. VLPs purified by ultracentrifugation. The blot is representative of one independent experiment conducted in triplicate. Quantification data are in Fig 5D.
(TIF)

## Acknowledgments

We thank members of the J.A.D. and M.O. laboratories for helpful discussions

## Author Contributions

**Conceptualization:** Abdullah M. Syed, Taha Y. Taha, Jennifer A. Doudna.

**Data curation:** Abdullah M. Syed, Alison Ciling, Irene P. Chen, Christopher R. Carlson, Armin N. Adly, Hannah S. Martin, Taha Y. Taha, Mir M. Khalid.

**Formal analysis:** Abdullah M. Syed.

**Funding acquisition:** David O. Morgan, Jennifer A. Doudna.

**Investigation:** Abdullah M. Syed, Alison Ciling, Irene P. Chen, Christopher R. Carlson, Armin N. Adly, Hannah S. Martin, Taha Y. Taha, Mir M. Khalid, Nathan Price, Mehdi Bouhaddou, Manisha R. Ummadi, Jack M. Moen.

**Methodology:** Abdullah M. Syed, Alison Ciling.

**Resources:** Jennifer A. Doudna.

**Supervision:** Nevan J. Krogan, David O. Morgan, Melanie Ott, Jennifer A. Doudna.

**Validation:** Abdullah M. Syed.

**Visualization:** Abdullah M. Syed.

**Writing – original draft:** Abdullah M. Syed, Alison Ciling.

**Writing – review & editing:** Abdullah M. Syed, Alison Ciling, Taha Y. Taha, David O. Morgan, Melanie Ott, Jennifer A. Doudna.

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
