## [Decision Letter · Decision Letter 0]

11 Jun 2024

Dear Dr. Doudna,

Thank you very much for submitting your manuscript "SARS-CoV-2 evolution balances conflicting roles of N protein phosphorylation" for consideration at PLOS Pathogens. As with all papers reviewed by the journal, your manuscript was reviewed by members of the editorial board and by several independent reviewers. In light of the reviews (below this email), we would like to invite the resubmission of a significantly-revised version that takes into account the reviewers' comments.

We cannot make any decision about publication until we have seen the revised manuscript and your response to the reviewers' comments. Your revised manuscript is also likely to be sent to reviewers for further evaluation.

Sincerely,

Justin Jang Hann Chu

Guest Editor

PLOS Pathogens

Ashley St. John

Section Editor

PLOS Pathogens

Michael Malim

Editor-in-Chief

PLOS Pathogens

orcid.org/0000-0002-7699-2064

Reviewer's Responses to Questions

**Part I - Summary**

Reviewer #1: Syed et al study the role of the N protein in the enhanced fitness of SARS-CoV-2 variants. They used single-cycle and virus-like particle assays to show that N phosphorylation in the central region of the protein has opposing effects on viral assembly and genome replication. The N protein from ancestral SARS-CoV-2 is highly phosphorylated, associated to high levels of genome replication but lower particle assembly compared to recent lineages. The authors also report on a new open reading frame, encoding a truncated N protein called N*, that occurs in the B.1.1 lineage and appears to support high levels of assembly and replication. N* improves viral fitness independently of phosphorylation. The results help explain the enhanced fitness of variants of concern.

Reviewer #2: This manuscript by Syed and colleagues describes the conflicting role of SARS-CoV-2 N protein phosphorylation in viral particle assembly and genome replication. It is an interesting study that provides insight into the pathogenesis of SARS-CoV-2.

Reviewer #3: In this manuscript, Doudna and colleagues describe the impact of SARS-CoV-2 nucleocapsid (N) protein phosphorylation and mutation of the SR motif within the linker region found in variants on viral assembly and replication. The authors first assessed recombinant protein phosphorylation using 32P-�-ATP labeling and SDS-PAGE and show a decrease in phosphorylation for N proteins containing previously described mutations P199L, S202R, R203M, and R203K. They next test the impact of these mutations on viral assembly using a VLP luciferase assay and treatment with different GSK-3 or kinase inhibitors. The data show that reduced phosphorylation results in increased assembly and inhibition of GSK activity recovered VLP assembly. To test the impact of phosphorylation on replication, the authors developed a single cycle infection assay for SARS-CoV-2 and generated �N SCIPs and found that increased N phosphorylation results in increased infection and growth rates that they relate to viral fitness. In addition, the authors evaluate the role of a low abundant, truncated N* protein that is found in the B.1.1. lineage and find that expression of N* appears to enhance viral assembly of VLPs and improve viral fitness. Overall, the studies described here appear to extend findings on the effect of N protein phosphorylation on SARS-CoV-2 replication and assembly. The coauthors previously analyzed the P199L, S202R, R203M, and R203K mutations present in variants of concern in a VLP assay and demonstrated an increase in VLP assembly (PMID: 34735219). However, two other studies also characterized the impact of enhanced phosphorylation of N protein with the evaluation of the S202R and R203K/G204R mutants (PMID: 36572560 and 35728038). They show that R203K/G204R increases phosphorylation of N and a corresponding increase in viral replication. While the authors here acknowledge that their findings are different, they do not provide additional rigorous experimental evidence to further bolster their model for N phosphorylation in replication. GSK-3 inhibition has also been shown in prior studies to impact N phosphorylation and the addition of other GSK-3 inhibitors in this study is complicated by effects of cytotoxicity. The one potentially interesting description here is the evaluation of a role for N* in assembly; however, studies here are limited by a lack of in-depth assessment of N*, including expression levels that would be relevant in infection and N* on the multiple roles N has in SARS-CoV-2 infection.

**Part II – Major Issues: Key Experiments Required for Acceptance**

Reviewer #1: 1.The abstract is quite vague regarding the main findings, and their importance regarding the recent variants.

2.The main text is also imprecise regarding which variants were used. For instance, page 4 bottom “we expressed and purified four SARS-CoV-2 N protein variants shown to enhance viral assembly”. The authors should indicate more clearly which variants they tested. They could also validate some of the results with recent variants. How is phosphorylated the N protein from the JN.1 lineage?

3.The authors show the phosphorylation state of N in transfected cells. Is it similar to that observed in infected cells?

Reviewer #2: 1. Is the observation consistent with other Sarbecoviruses, such as those detected in bats and pangolins, and also betacoronaviruses, such as MERS-CoV? Including at least other sarbecoviruses could significantly improve the significance of the current study. Is the degree of phosphorylation in N protein a factor contributing to the differences in disease severity between SARS-CoV-2 and SARS-CoV-1?

2. I wondered why the author generated a spike-competent, N-deleted replicon system instead of both spike and N-deleted replicon system. A spike-deficient replicon system will only replicate a single round and might be the best model to study replication.

3. What are the underlying factors of not including omicron variants (R203K + G204R) in your fitness study? I suggest including this as two recent studies (As highlighted in your discussion) demonstrated a different observation.

4. I was wondering why there is no increase in viral titers in Calu cells after R203M or S203R substitution at the N of the WA1 infectious clone. This work mainly uses cell lines and might not represent actual viral pathogenesis in vivo. It might be good to validate the results with organoid systems, or with animal model. 

5. N phosphorylation could be one of the factors but might not be the only factor determining the replication fitness of the SARS-related coronaviruses. Other factors, such as receptor usage, entry efficiency, and degree of immune evasiveness, could also determine the risk of pre-pandemic SARS-related coronavirus.

Reviewer #3: It is surprising that statistical analyses are lacking for most subfigures. The number of independent replicates is also not stated for most experiments, so it is not clear how rigorous the experiments are for these studies. For Fig. 4D, n=1 is not sufficient.

Two other studies also characterized the impact of enhanced phosphorylation of N protein with the evaluation of the S202R and R203K/G204R mutants (PMID: 36572560 and 35728038). Findings here shown in Fig. 1 is opposite for some of their mutants. Stating that, “Both studies used PhosTag gels to evaluate phosphorylation, but this approach produced broad, difficult to interpret smears in our experiments due to the high degree of phosphorylation in N protein. The mutation at S202R in particular is unlikely to increase phosphorylation since it should block GSK3 phosphorylation at sites 202, 198, 194, 190 and 186” does not provide a satisfactory explanation or sufficient evidence. Orthogonal approaches to assess and confirm phosphorylation should be performed given the discrepancy of results.

Fig. 2C, the authors make the comparison to SARS-CoV-1 N. How conserved is the linker sequence between CoV-2 N and CoV-1 N? Why was the comparison done with CHIR98014 and not the more specific SB216763 as in Fig. 2B?

It is not clear how chimeric N protein variants generated by swapping sequences with MHV were validated and shown not to be “interfering with other N protein functions”. In this study, a luciferase readout was used to assess their impact in viral assembly and replication. No additional evaluation of “other vital N protein functions” were performed.

The authors developed the SCIP system but did not use it to test their N mutants of interest. Why? If the system uncouples entry from replication, it would have strengthened their results to see assays with trans-complemented mutant N proteins.

Fig. 3B, what is the rationale for using CHIR98014 and not SB216763? Based on Fig. S2, viability significantly decreases at CHI98014 concentrations >1 uM in 293T cells. This may be reflected in other cell types shown in Fig. 3D-H (cell viability for those cell types are not shown) and not reduced replication.

Evaluation of a role for N* is potentially interesting. However, the studies here are very limited. Fig. 5C does not show protein expression levels of truncated N-variants; lack of stable expression can account for lack of luciferase activity in packaging. What is the N* expression levels in VLPs? Western blot data should be shown along with the quantitation. With the VLPs, it is not clear what the N* expression levels are relative to N wt. Studies here do not examine the impact of N* on the multiple roles of N in viral replication and pathogenesis.

**Part III – Minor Issues: Editorial and Data Presentation Modifications**

Reviewer #1: (No Response)

Reviewer #2: 1. Be consistent with the graph y-axis, and it might be good to display the graph in a log scale.

Reviewer #3: Fig 3B legend, missing description.

Fig. S3, times on images can be made clearer.

Fig. 4E-G, no drug data can be made darker.

PLOS authors have the option to publish the peer review history of their article (what does this mean?). If published, this will include your full peer review and any attached files.

Reviewer #1: No

Reviewer #2: No

Reviewer #3: No
---

## [Decision Letter · Decision Letter 1]

11 Nov 2024

Dear Dr. Doudna,

We are pleased to inform you that your manuscript 'SARS-CoV-2 evolution balances conflicting roles of N protein phosphorylation' has been provisionally accepted for publication in PLOS Pathogens.

Best regards,

Justin Jang Hann Chu

Guest Editor

PLOS Pathogens

Ashley St. John

Section Editor

PLOS Pathogens

Michael Malim

Editor-in-Chief

PLOS Pathogens

orcid.org/0000-0002-7699-2064

The authors have addressed the concerns of the reviewers adequately.

Reviewer Comments (if any, and for reference):

Reviewer's Responses to Questions

**Part I - Summary**

Reviewer #2: NA

Reviewer #3: In this revised manuscript, Syed and colleagues have adequately addressed most concerns raised by the reviewers to describe the role of N protein phosphorylation in SARS-CoV-2 assembly, replication, and viral fitness. These changes have improved the overall manuscript.

**Part II – Major Issues: Key Experiments Required for Acceptance**

Reviewer #2: The authors have adequately addressed my concerns.

Reviewer #3: (No Response)

**Part III – Minor Issues: Editorial and Data Presentation Modifications**

Reviewer #2: (No Response)

Reviewer #3: (No Response)

PLOS authors have the option to publish the peer review history of their article (what does this mean?). If published, this will include your full peer review and any attached files.

Reviewer #2: No

Reviewer #3: No

---

## [Editor Report · Acceptance letter]

14 Nov 2024

Dear Dr. Doudna,

We are delighted to inform you that your manuscript, "SARS-CoV-2 evolution balances conflicting roles of N protein phosphorylation," has been formally accepted for publication in PLOS Pathogens.

Best regards,

Michael Malim

Editor-in-Chief

PLOS Pathogens

orcid.org/0000-0002-7699-2064